# A Two-Branch Convolutional Neural Network Based on Multi-Spectral Entropy Rate Superpixel Segmentation for Hyperspectral Image Classification

**Caihong Mu** [1], **Zhidong Dong** [1] and **Yi Liu** [2,*]

1    Key Laboratory of Intelligent Perception and Image Understanding of Ministry of Education, Collaborative Innovation Center of Quantum Information of Shaanxi Province, International Research Center for Intelligent Perception and Computation, Joint International Research Laboratory of Intelligent Perception and Computation, School of Artificial Intelligence, Xidian University, Xi'an 710071, China; caihongm@mail.xidian.edu.cn (C.M.); zhidong_dong@stu.xidian.edu.cn (Z.D.)

2    School of Electronic Engineering, Xidian University, Xi'an 710071, China

*    Correspondence: yiliu@xidian.edu.cn

**Abstract:** Convolutional neural networks (CNNs) can extract advanced features of joint spectral–spatial information, which are useful for hyperspectral image (HSI) classification. However, the patch-based neighborhoods of samples with fixed sizes are usually used as the input of the CNNs, which cannot dig out the homogeneousness between the pixels within and outside of the patch. In addition, the spatial features are quite different in different spectral bands, which are not fully utilized by the existing methods. In this paper, a two-branch convolutional neural network based on multi-spectral entropy rate superpixel segmentation (TBN-MERS) is designed for HSI classification. Firstly, entropy rate superpixel (ERS) segmentation is performed on the image of each spectral band in an HSI, respectively. The segmented images obtained are stacked band by band, called multi-spectral entropy rate superpixel segmentation image (MERSI), and then preprocessed to serve as the input of one branch in TBN-MERS. The preprocessed HSI is used as the input of the other branch in TBN-MERS. TBN-MERS extracts features from both the HSI and the MERSI and then utilizes the fused spectral–spatial features for the classification of HSIs. TBN-MERS makes full use of the joint spectral–spatial information of HSIs at the scale of superpixels and the scale of neighborhood. Therefore, it achieves excellent performance in the classification of HSIs. Experimental results on four datasets demonstrate that the proposed TBN-MERS can effectively extract features from HSIs and significantly outperforms some state-of-the-art methods with a few training samples.

**Keywords:** hyperspectral image classification; two-branch neural network; multi-spectral entropy rate superpixel segmentation

## 1. Introduction

Hyperspectral images (HSIs) are obtained by simultaneously imaging ground objects in a certain area on continuous multiple spectral bands. The classification of HSIs is an important technology in remote sensing, as HSIs contain rich spatial and spectral information [1–3], which is widely applied in the fields of military target recognition, geological resource detection, agricultural crop monitoring, and archaeological relic restoration [4–7]. HSIs have a very high spectral dimension and rich spectral information, but the spectral information is mixed with noise information brought by atmospheric environment and imaging instruments. The classification accuracy of HSIs is often not good enough if the original information is directly used. Effective feature extraction of HSIs is one of the important means to improve classification accuracy [8–10]. In addition, due to the high cost of manual labeling, the labels of HSIs are usually limited. How to achieve higher classification accuracy with fewer labeled samples is an important direction in the research of HSI classification [11].

In recent decades, the classification of HSIs has received more and more attention. Researchers have made considerable progress [12–14]. Only the spectral information of HSI was used for classification in early research, such as K-nearest neighbors (KNN) [15], support vector machine (SVM) [16,17], sparse representation [18], and so on. These methods classified samples according to the spectral feature and were easy to implement, but the classification accuracy was relatively limited. Subsequently, some methods [19,20] combining spatial and spectral information were proposed, which led to a better classification performance. However, due to the high dimension of HSIs and the intra-class differences of ground objects, how to obtain better classification accuracy is still a difficult problem that needs to be investigated further.

With the vigorous development of deep learning, researchers began to apply deep learning to HSI classification. A lot of methods were proposed such as the stacked auto-encoder (SAE) [19], deep belief networks (DBNs) [20], and convolutional neural networks (CNNs) [21–23]. Among all these methods, CNN-based methods are the most widely used. CNN-based methods take the patch-based neighborhoods of samples as input and perform feature extraction through convolution. The spatial features and the spectral features, or the joint features of spatial and spectral information can be extracted by convolution from the patch-based neighborhoods, and the classification accuracy of HSIs is greatly improved. Hamida et al. [24] studied the effect of different three-dimensional (3D) convolutional networks on HSI classification and designed a 3DCNN network consisting of 4 layers of 3D convolution and a fully connected layer to extract the spectral–spatial features from HSIs. Hamida's 3DCNN model made good use of joint features and achieved a higher classification accuracy. Zhong et al. [25] proposed a spectral–spatial residual network (SSRN), in which they designed a spatial feature extraction module and a spectral feature extraction module using multiple 3D convolutional layers according to the residual structure and investigated the effects of number and order of modules in feature extraction. This end-to-end feature extraction and classification network of HSIs achieved high classification accuracy. Roy et al. [26] proposed a hybrid spectral convolutional neural network (HybridSN), which firstly used multiple 3D convolutional layers for spectral–spatial feature extraction, and then used a two-dimensional (2D) convolutional layer to extract advanced spatial feature from the obtained feature maps. The 2D convolutional layer reduced the complexity of the model and improved the classification accuracy.

The CNN-based methods mostly take the patch-based neighborhoods of samples as input, which means that the spatial information only comes from the neighborhood, and the spatial information outside the neighborhood is ignored. Some researchers tried to propose some methods that utilized a wider area of spatial information. Superpixel segmentation is a commonly used means to obtain spatial distribution information of ground objects in a wide area. Leng et al. [27] used entropy rate superpixel (ERS) segmentation to perform multi-scale superpixel segmentation on the first principal component of HSIs and further extracted the spectral–spatial information within each superpixel for classification. Jiang et al. [28] proposed a superpixel-wise principal component analysis (PCA) approach for unsupervised feature extraction of HSIs (SuperPCA). SuperPCA firstly performed multi-scale ERS on the first principal component of HSIs, and then applied the proposed superpixel-wise principal component analysis approach to reduce the dimension of HSIs within the obtained superpixels. Finally, it trained classifiers over the data after dimension reduction on each scale, respectively, and obtained the results through decision fusion. This method achieved higher classification accuracy even when the training samples were limited.

A two-branch network is a commonly used structure to process multi-source data or extract a variety of different features [29–32]. The two branches may have different structures or same structures with different parameters. According to the different inputs of the two branches, each branch processes a certain type of information. Feature fusion will usually be performed at the end of network. The spectral–spatial attention network proposed by Mei et al. [29] is a two-branch network. One branch uses the structure of recur-

rent neural network to extract spectral information, the other branch uses CNN structures to extract spatial information, and an attention mechanism is applied to make the network able to extract key fusion features. Mu et al. [30] proposed a low-rank based method for HSI classification called a two-branch network combined with robust PCA, where both the low rank and the sparse components were preserved and used for feature extraction in two independent convolutional branches. This method constructed a convenient model for HSI classification by discarding the low-rank subspace estimation and combining denoising, feature extraction, feature fusion, and classification into an end-to-end network, which maintained better classification performance even for the cases of small samples and class imbalance.

The above methods have made great progress in improving the classification results of HSIs, but some issues still need to be addressed. These methods mainly use the rich spectral features of HSIs and extract joint spectral–spatial features from the patch-based neighborhoods of the samples. In fact, the information of each band of HSIs is quite different, and each band has rich spatial information that has not been fully utilized. If the various bands of the HSI are processed separately, more representative spatial information can be extracted. On the other hand, the pixels corresponding to the same ground object, which may distribute in a wide area, should belong to the same category. Mining and making use of such priori knowledge can improve the classification results of HSIs further. In high-resolution images, the same kind of ground objects are distributed in a continuous area with an irregular shape, and the size is much larger than that of the patch-based neighborhoods. When the classifier just takes the patch-based neighborhoods as input, it can only extract features in the scale of patch-based neighborhoods, and the spatial distribution information of the ground objects outside the patches is ignored.

To solve the above problems, this paper designs a two-branch convolutional neural network based on multi-spectral entropy rate superpixel segmentation (TBN-MERS) for HSI classification. By performing ERS segmentation on each band of HSIs, TBN-MERS can divide a certain area of the same kind of ground objects into a same superpixel, so as to obtain the superpixel-scale spatial information of each band. The preprocessed HSI and the preprocessed multi-spectral ERS image (MERSI) are then used as the two inputs of the two-branch convolutional neural network, which can extract the joint information of the spectral–spatial features at the scale of both the patch-based neighborhoods and the superpixels.

The main contributions of this paper are as follows. (1) ERS segmentation is performed on all bands of HSIs, and the obtained MERSI contains rich superpixel-scale spatial information, which greatly improves the classification accuracy of HSIs. (2) A two-branch convolutional neural network is designed, which can effectively extract and fuse the preprocessed HSI and preprocessed MERSI, and the fusion feature further improves the classification accuracy. (3) TBN-MERS can obtain much better classification results, even with very limited samples.

The rest of this paper is as follows. Section 2 introduces the related backgrounds and the details of TBN-MERS. Section 3 first analyzes the effect of the structure and the parameters in TBN-MERS, then describes the comparison experiments between TBN-MERS and its variants, and finally, compares TBN-MERS with other state-of-the-art methods on four datasets. The last section summarizes the work of this paper and provides the direction of the further work in the future.

## 2. Materials and Methods

### 2.1. Multi-Spectral ERS

Superpixel segmentation is a kind of commonly used preprocessing method in computer vision fields such as target detection and image segmentation [33–35]. Superpixel segmentation is to divide the pixels in the image into different groups with certain properties. The pixels belonging to the same superpixel are similar in texture, brightness, color, or other characteristics [36]. The purpose of superpixel segmentation is to achieve the

following effects. (a) Each superpixel contains only one class of ground object. (b) The set of superpixel boundaries is a super set of the boundaries of ground objects [37].

The methods of generating superpixels can be categorized into clustering-based methods and graph-based methods. Clustering-based methods start from a rough initial clustering of pixels, and iteratively refine the clustering to form superpixels until certain convergence criteria are met, such as Mean Shift [38], Watershed [39], and Simple Linear Iterative Clustering (SLIC) [36]. The graph-based methods regard each pixel in the image as a node of the graph. The weight of edges between two nodes is calculated based on the similarity of them. Graph-based methods include Normalized Cuts [40], Felzenszwalb [41], Entropy Rate Superpixel Segmentation (ERS) [37], and so on.

Mean Shift [38] is an iterative method to find the pattern among pixels in the feature space by maximizing the local density function. The pixels that converge to the same pattern form a superpixel. Watershed [39] regards the image as a geographical topological structure, and the value of a pixel in the image represents the altitude. The connection of the pixels with large values is regarded as a ridge, the area formed by the pixels with small values is regarded as a valley, and the closed valley is regarded as a superpixel. SLIC [36] maps the image to a new feature space, clusters it according to the distance metric, and regards different clusters as superpixels. Normalized Cuts [40] uses contour and texture to recursively segment the image and minimizes the global cost function defined on the segmentation boundary. Felzenszwalb [41] regards the image as graph and clusters pixels on the graph, so that each superpixel is the smallest spanning tree of pixels. ERS [37] finds multiple disconnected sub-graphs from the graph of the image by iteratively optimizing the entropy rate loss function, and each sub-graph represents a superpixel.

Mean Shift is robust to local differences but tends to straddle multiple objects. Watershed runs very quickly, but the research of Levinshtein et al. [42] and Veksler et al. [43] show that the superpixels produced by Watershed often contain multiple object categories. SLIC has excellent performance in efficiency and segmentation effect, and is one of the commonly used segmentation algorithms. Normalized Cuts generates superpixels with the uniform size and compact shape, but it requires a lot of calculations and runs slowly. Felzenszwalb is very efficient and generates a good embedding representation. However, Ren et al. [34] proved that it tended to sacrifice details so that it generated smooth boundaries, and the boundary recall rate was not high enough. ERS has high efficiency and high boundary recall rate, and the obtained superpixels are compact and homogeneous. In this paper, ERS is applied for superpixel segmentation.

ERS regards superpixel segmentation as an undirected graph clustering problem [37]. The image is represented by a graph $G' = (V, E)$, in which $V$ is a set of all pixels, and $E$ is the set of edges that are between every pixel and its adjacent pixels. The purpose of ERS is to find a sub-graph $G' = (V, A)$, $A \subseteq E$. $G' = (V, A)$ represents the image after superpixel segmentation. The objective function of ERS includes two parts: (1) the entropy rate $H(A)$ calculated by random walk on the graph; and (2) the balance term $B(A)$ of the clustering distribution. In the process of superpixel segmentation, a larger entropy rate $H(A)$ will make the obtained superpixels more compact and homogeneous, which is conducive to containing only a single ground object in each superpixel, and the balance item $B(A)$ controls the size of the cluster, avoiding it becoming too smooth while retaining the boundary of ground object. The objective function of ERS is as follows:

$$A^* = argmax(H(A) + \alpha B(A)), \quad s.t. A \subseteq E \tag{1}$$

where $\alpha$ is a trade-off parameter of $H(A)$ and $B(A)$.

ERS has been applied to a variety of HSI classification methods to extract spatial features, but the existing methods generally only perform ERS on the first principal component of HSIs and do not consider the spatial differences between different bands. To fully utilize the spatial information of HSIs in different bands, and to correlate the spatial information with spectral information better, we perform ERS on images of all bands in HSIs.

Taking the Pavia University (PU) dataset as the input, the process of multi-spectral ERS of HSIs proposed in this paper is shown in Figure 1. First, the HSI is separated according to bands, and then the values in the image of each band are scaled to the interval [0, 255]. Suppose $X$ represents the set of all pixels of the image in a certain band, $x_i$ represents the value of certain pixel in $X$, $Z(x_i)$ is the corresponding value after scaling, $min(X)$ is the minimum value of $X$, and $max(X)$ is the maximum value of $X$, then the scaling formula is:

$$Z(x_i) = \frac{x_i - min(X)}{max(X) - min(X)} \times 255 \qquad (2)$$

We perform ERS on the scaled image of each band to obtain the superpixel segmentation image of each band. ERS will divide the pixels into $n_c$ superpixels. The superpixels are numbered $1, 2, \ldots, n_c$ respectively. In the obtained segmentation image, the values of all pixels in the same superpixel are all equal to the serial number of the superpixel. Finally, the segmentation images are stacked one by one according to order of the bands to form the MERSI.

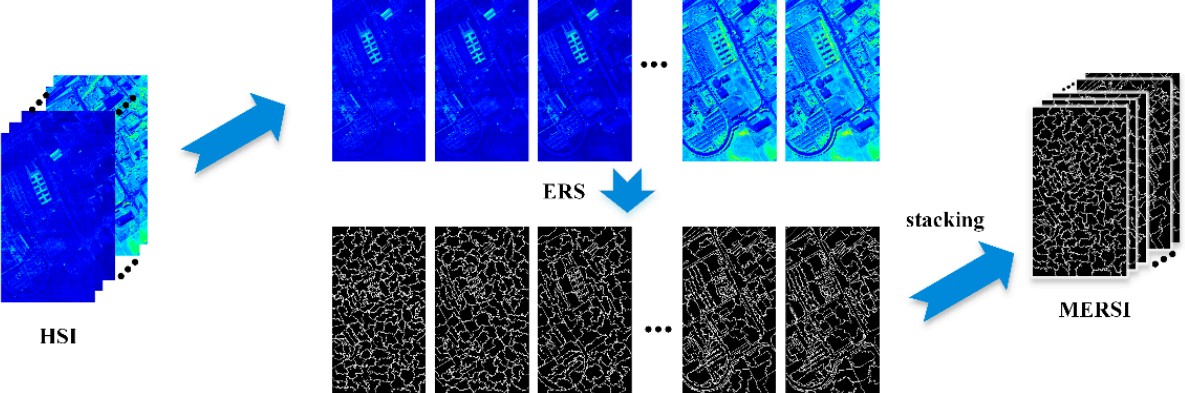

**Figure 1.** The process of the multi-spectral entropy rate superpixel segmentation.

### 2.2. Two-Branch Convolutional Neural Network (TBN-MERS)

CNNs are mainly composed of convolutional layers. The convolutional layer uses multiple convolution kernels to perform convolution on the multi-spectral feature map input to obtain a more advanced feature map. The convolutional layer has the characteristics of local connection and weight sharing; that is, each node of the output is only connected to some nodes of the input, and the weights of connections between the nodes in different positions in output and corresponding nodes in input are the same. The convolution operation can keep roughly the same spatial structure of the output data with that of the input data, so it is widely used in the feature extraction of two-dimensional (2D) and three-dimensional (3D) images. Commonly used convolutional layers include 2D convolutional layers and 3D convolutional layers. The 2D convolutional layer mainly performs feature extraction from the spatial dimension, and the 3D convolutional layer can perform feature extraction from the spatial dimension and the spectral dimension at the same time. We use 2D convolutional layers, 3D convolutional layers, and fully connected layers to build neural networks. The HybridSN [26] proposed by Roy et al. is an end-to-end single-branch network, which is very effective in the feature extraction of HSIs. It has achieved high classification accuracy on multiple datasets. We design a two-branch convolutional neural network (TBN-MERS) based on the structure of HybridSN.

Taking the PU dataset as the input, Figure 2 shows the network structure of TBN-MERS. TBN-MERS has two feature extraction branches, with the same structure, where 3 layers of 3D convolutional layer and 1 layer of 2D convolutional layer are cascaded, and after each 3D convolutional layer and 2D convolutional layer, there is a BatchNorm layer and a ReLU activation layer. The outputs of the two network branches are then flattened

as vectors and are added by the element addition operation to implement feature fusion. Subsequently, 3 cascaded fully connected layers are used for classification, where one dropout layer follows each of the first two fully connected layers. Finally, the Softmax function is applied to convert the network output into a probability vector. In Figure 2, we use Conv3D to represent the cascade of a 3D convolutional layer, a BatchNorm layer, and a ReLU layer, use Conv2D to represent the cascade of a 2D convolutional layer, a BatchNorm layer, and a ReLU layer, use FC to represent the cascade of a fully connected layer and a Dropout layer, and use fc represents the cascade of a fully connected layer and a Softmax layer. The hyper-parameters in the two-branch neural network are shown in Table 1. In Table 1, Conv3D_1, Conv3D_2, and Conv3D_3 represent different Conv3Ds that are cascaded in two branches of the same structure, and FC_1, FC_2 represent different FCs that are cascaded in the network.

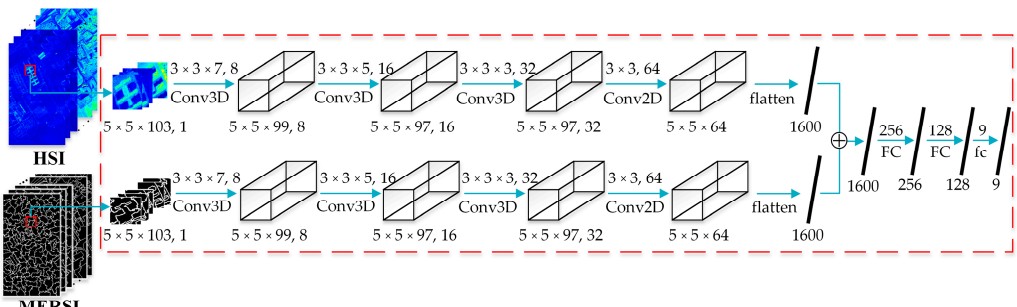

**Figure 2.** The network structure of TBN-MERS (Conv3D represents the cascade of 3D convolutional layer, BatchNorm layer and ReLU layer; Conv2D represents the cascade of 2D convolutional layer, BatchNorm layer and ReLU layer; FC represents the cascade of fully connected layer and dropout layer; fc represents the cascade of the fully connected layer and the Softmax layer).

**Table 1.** The hyper-parameters of the TBN-MERS.

| Layers | Kernel Size | Number of Kernels | Padding |
|---|---|---|---|
| Cov3D_1 | (3,3,7) | 8 | 1 |
| Cov3D_2 | (3,3,5) | 16 | 1 |
| Cov3D_3 | (3,3,3) | 32 | 1 |
| Cov2D | (3,3) | 64 | 1 |
| FC_1 | – | 256 | – |
| FC_2 | – | 128 | – |
| fc | – | Number of classes | – |

The BatchNorm layer standardizes the input batch data and utilizes scaling variables $\gamma$ and translation variables $\beta$ to adjust the mean and variance of the standardized batch data in order to get a better distribution of values. Let $\mathbf{B} = \{x_1, x_2, \ldots, x_m\}$ represent a batch of data and $BN(x_i)$ represent the corresponding output of sample $x_i$ after BatchNorm layer; then, the BatchNorm operation is as follows:

$$BN(x_i) = \gamma \hat{x}_i + \beta \tag{3}$$

In Formula (3), $\hat{x}_i = \frac{x_i - \mu_B}{\sqrt{\sigma_B^2 + \varepsilon}}$, where $\sigma_B^2 = \frac{1}{m} \sum_{i=1}^{m} (x_i - \mu_B)^2$, $\mu_B = \frac{1}{m} \sum_{i=1}^{m} x_i$, and $\varepsilon$ is a very small number to stabilize the value.

The ReLU activation function can introduce nonlinearity to the neural network and make the gradient propagate more efficiently. For the input data, the ReLU activation function can be expressed as:

$$ReLU(x_i) = \begin{cases} x_i, x_i > 0 \\ 0, x_i \leq 0 \end{cases} \tag{4}$$

### 2.3. The Process of TBN-MERS

The overall process of proposed TBN-MERS is shown in Figure 3, and the details are described as follows.

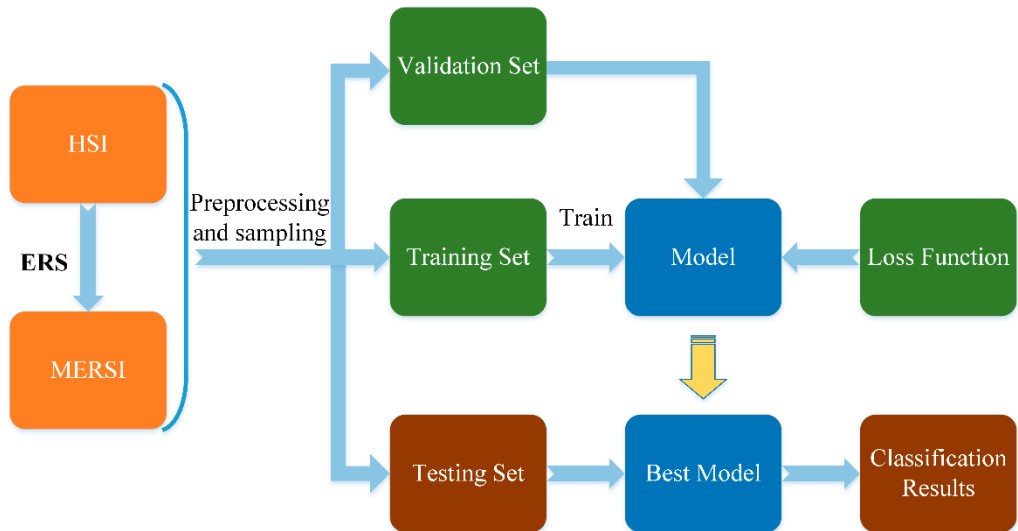

**Figure 3.** The overall process of TBN-MERS.

The data cube of one HSI is represented as $\mathbf{X} \in \mathbb{R}^{w \times h \times b}$, where $w$ is the width and $h$ is the height of the image, and $b$ is the number of spectral bands. Let $\mathbf{Y} \in \mathbb{R}^{w \times h}$ represent the corresponding label image, and the value $y_{ij}$ at the location $(i, j)$ of the label map $\mathbf{Y}$ is chosen from the set $\{0, 1, 2, \ldots, c\}$, where $c$ represents the total number of classes.

$\mathbf{X}_k \in \mathbb{R}^{w \times h}, k \in [1, b]$ represents the image in the $k$th band. The value of all pixels on $\mathbf{X}_k$ is scaled to the interval $[0, 255]$, and ERS algorithm is applied to obtain the corresponding segmentation image $\mathbf{S}_k$. After obtaining the segmentation images of all spectral bands, we stack them according to the order of the bands to obtain the data cube of superpixel segmentation image $\mathbf{S} \in \mathbb{R}^{w \times h \times b}$ corresponding to $\mathbf{X}$.

Firstly, the HSI data $\mathbf{X}$ are preprocessed, and the obtained data cube is denoted by $\mathbf{X}'$. The preprocessing operation is to standardize the image of each band. Let $x_i$ represent the value of a certain pixel in $\mathbf{X}_k$, and let $Standardize(x_i)$ represent the value of this pixel after standardization; then, the standardization formula is:

$$Standardize(x_i) = \frac{x_i - \mu_k}{\sigma_k} \tag{5}$$

where $\mu_k$ is the mean, and $\sigma_k$ is the standard deviation of all pixel values in $\mathbf{X}_k$.

Secondly, the MERSI data $\mathbf{S}$ is preprocessed, and the obtained data cube is denoted by $\mathbf{S}'$. The preprocessing operation is to normalize the image of each band. Let $s_i$ represent the value of a certain pixel in $\mathbf{S}_k$, $Normalize(s_i)$ represent the value of this pixel after normalization, $min(\mathbf{S}_k)$ represent the minimum value of $\mathbf{S}_k$, and $max(\mathbf{S}_k)$ represent the maximum value of $\mathbf{S}_k$; then, the normalization formula is:

$$Normalize(s_i) = \frac{s_i - min(\mathbf{S}_k)}{max(\mathbf{S}_k) - min(\mathbf{S}_k)} \tag{6}$$

Thirdly, we generate the overall sample set. Each sample is corresponding to a pixel in the HSI and consists of two parts: the patch-based neighborhood $p_i^x \in \mathbb{R}^{p \times p \times b}$ taken from the preprocessed data cube $\mathbf{X}'$, where $p$ is the height or width of the patch and $b$ is the number of the bands in the HSI; the patch-based neighborhood $p_i^s \in \mathbb{R}^{p \times p \times b}$ taken from the preprocessed data cube $\mathbf{S}'$. We randomly select a certain number of samples from each class to form the training set and draw an equal number of samples from the remaining

samples of each class to form the validation set, and the remaining samples are used as the test set.

Finally, we feed the two parts $p_i^x$ and $p_i^s$ of each sample $x_i$ into the two branches of TBN-MERS and train the network. We test the model on the verification set every epoch. When the classification accuracy of the model on the validation set no longer rises, the training is completed. We use the best model on the validation set to predict the samples in testing set.

The cross-entropy loss function is used to train the network. The cross-entropy loss function is as follows:

$$L = -\frac{1}{N}\sum_i^N \sum_j^c y_{ij} \log(p_{ij}) \tag{7}$$

where $N$ is the number of samples in a batch, $x_i$ is the $i$th sample in the batch, $c$ is the number of classes, and $p_{ij}$ is the probability value that the sample $x_i$ belongs to the class j predicted by the model. When the sample $x_i$ belongs to the class $j$, the value of $y_{ij}$ is 1; otherwise, it is 0.

The overall process of TBN-MERS is shown in Figure 3.

## 3. Results

### 3.1. Datasets

We conducted experiments on the Indian Pines (IP), Pavia University (PU), Salinas (SA), and Houston (HU) datasets.

IP was captured by the American AVIRIS sensor in the Indian remote sensing experimental area with a spatial resolution of 20 m. The AVIRIS sensor divided the spectrum from 375 μm to 2200 μm into 220 bands. After removing the 20 bands contaminated by noise, the remaining 200 spectral bands were retained. IP included 16 classes of ground objects, including grass pasture, woods, and wheat, etc. The image size of IP was $145 \times 145$. After removing the background pixels in the image, a total of 10,249 labeled samples could be used for classification. Figure 4a is the true label image of IP. Table 2 lists the specific classes of IP and the number of labeled samples of each class.

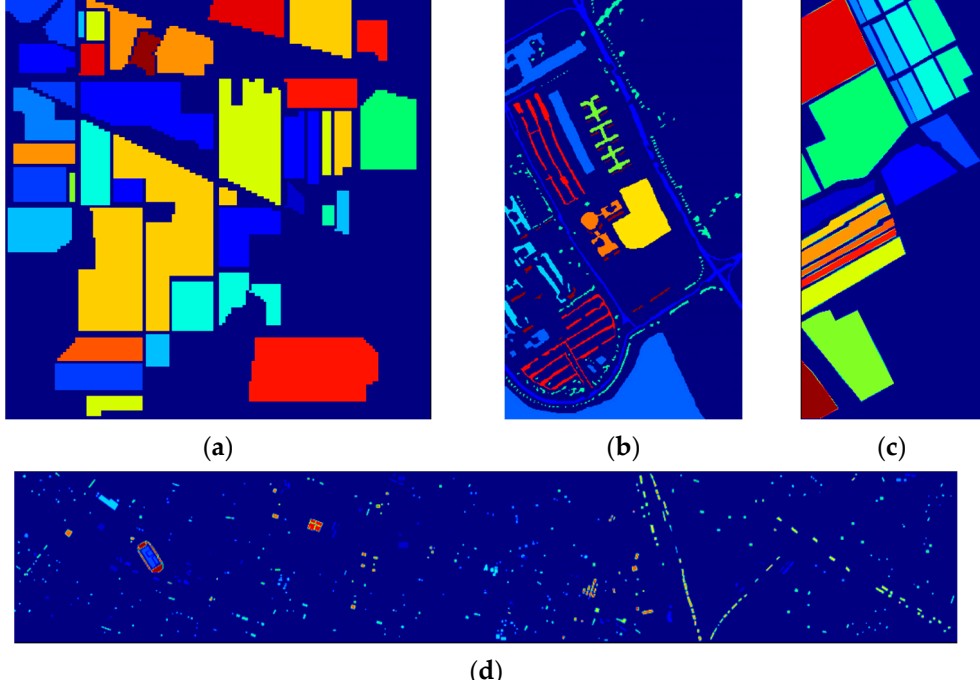

**Figure 4.** True label images of the four datasets (**a**) IP, (**b**) PU, (**c**) SA, and (**d**) HU.

**Table 2.** The information of each class in IP.

| Number | Class | Samples | Color |
|---|---|---|---|
| 1 | Alfalfa | 46 | |
| 2 | Corn-notill | 1428 | |
| 3 | Corn-mintill | 830 | |
| 4 | Corn | 237 | |
| 5 | Grass-pasture | 483 | |
| 6 | Grass-trees | 730 | |
| 7 | Grass-pasture-mowed | 28 | |
| 8 | Hay-windrowed | 478 | |
| 9 | Oats | 20 | |
| 10 | Soybean-notill | 972 | |
| 11 | Soybean-mintill | 2455 | |
| 12 | Soybean-clean | 593 | |
| 13 | Wheat | 205 | |
| 14 | Woods | 1265 | |
| 15 | Buildings-Grass-Trees-Drives | 386 | |
| 16 | Stone-Steel-Towers | 93 | |
| Total | | 10,249 | |

PU was captured by the ROSIS sensor at the University of Pavia in northeastern Italy with a spatial resolution of 1.3 m. The ROSIS sensor detected 115 bands in the wavelength range of 0.43 μm to 0.86 μm. After 12 noisy bands are removed, data from the remaining 103 bands are used for experiments. PU includes nine classes of ground objects, such as asphalt, meadows, gravel, etc. The image size of PU is 610 × 340. After removing background pixels in the image, a total of 42,776 labeled samples could be used for classification. Figure 4b is the true label image of PU. Table 3 lists the specific classes of PU and the number of labeled samples of each class.

**Table 3.** The information of each class in PU.

| Number | Class | Samples | Color |
|---|---|---|---|
| 1 | Asphalt | 6631 | |
| 2 | Meadows | 18,649 | |
| 3 | Gravel | 2099 | |
| 4 | Trees | 3064 | |
| 5 | Painted metal sheets | 1345 | |
| 6 | Bare Soil | 5029 | |
| 7 | Bitumen | 1330 | |
| 8 | Self-Blocking Bricks | 3682 | |
| 9 | Shadows | 947 | |
| Total | | 42,776 | |

SA was captured by the American AVIRIS sensor in the Salinas Valley in California. It had a spatial resolution of 3.7 m. This image had 224 bands originally. Removing the bands that could not be reflected by water, the remaining 204 bands were generally used for classification. SA included 16 classes of ground objects, such as fallow, celery, etc. It contained 111,104 pixels in total, of which 56,975 pixels were background pixels, and 54,129 pixels could be used for classification. The image size of SA was 512 × 217. Figure 4c is the true label image of SA. Table 4 lists the specific classes of SA and the number of labeled samples of each class.

**Table 4.** The information of each class in SA.

| Number | Class | Samples | Color |
|--------|-------|---------|-------|
| 1 | Brocoli_green_weeds_1 | 2009 | |
| 2 | Brocoli_green_weeds_2 | 3726 | |
| 3 | Fallow | 1976 | |
| 4 | Fallow_rough_plow | 1394 | |
| 5 | Fallow_smooth | 2678 | |
| 6 | Stubble | 3959 | |
| 7 | Celery | 3579 | |
| 8 | Grapes_untrained | 11,271 | |
| 9 | Soil_vinyard_develop | 6203 | |
| 10 | Corn_senesced_green_weeds | 3278 | |
| 11 | Lettuce_romaine_4wk | 1068 | |
| 12 | Lettuce_romaine_5wk | 1927 | |
| 13 | Lettuce_romaine_6wk | 916 | |
| 14 | Lettuce_romaine_7wk | 1070 | |
| 15 | Vinyard_untrained | 7268 | |
| 16 | Vinyard_vertical_trellis | 1807 | |
| Total | | 54,129 | |

HU was captured by the ITRES CASI-1500 sensor with a spatial resolution of 2.5 m, provided by the 2013 IEEE GRSS Data Fusion Contest. HU contained 144 bands ranging from 364 nm to 1046 nm. HU included 15 classes of ground objects such as trees, soil, residential, etc. After removing the background pixels in the image, a total of 15,268 labeled samples could be used for classification. Figure 4d is the true label image of HU. Table 5 lists the specific classes of HU and the number of labeled samples of each class.

**Table 5.** The information of each class in HU.

| Number | Class | Samples | Color |
|--------|-------|---------|-------|
| 1 | Healthy grass | 1251 | |
| 2 | Stressed grass | 1254 | |
| 3 | Synthetic grass | 732 | |
| 4 | Trees | 1244 | |
| 5 | Soil | 1242 | |
| 6 | Water | 339 | |
| 7 | Residential | 1268 | |
| 8 | Commercial | 1244 | |
| 9 | Road | 1252 | |
| 10 | Highway | 1227 | |
| 11 | Railway | 1288 | |
| 12 | Parking Lot 1 | 1233 | |
| 13 | Parking Lot 2 | 531 | |
| 14 | Tennis Court | 463 | |
| 15 | Running Track | 700 | |
| Total | | 15,268 | |

### 3.2. Experimental Settings

In order to analyze the effect of different factors on the classification performance, we conducted a series of experiments, including: (1) the difference between two-branch network and single-branch network; (2) the influence of the number of superpixels in ERS and the patch size of input samples; (3) the difference of multi-spectral methods based on two kinds of superpixel segmentation methods, ERS and SLIC; (4) the different effect of segmentation images obtained by applying ERS to images on multiple bands and the first principal component; and (5) a comparison between TBN-MERS and other methods.

In multi-spectral ERS, the number of superpixels in each band $n_c$ is set to 50 for IP and SA and 200 for PU and HU. In ERS, the trade-off parameter $\alpha$ is set to 0.5. For IP, PU, and HU datasets, 50 samples are randomly selected from each class (10 samples are selected

from the class with less than 50 samples) to form the training set. For SA, five samples are randomly selected from each class to form the training set. For each dataset, the number of randomly selected samples in the validation set is the same as that of the training set. The remaining samples are used to form the test set. In TBN-MERS, the training batch size is set to 32, and SGD optimizer is adopted. The learning rate is set to 0.0005. In SVM, only the spectral vector corresponding to the local center pixel is taken as the input. In TBN-MERS, Net-X, and Net-S (in Section 3.3.1), the patch size *p* of samples from IP, PU, and SA is set to 5, and the patch size *p* of samples from HU is set to 7. To achieve the best effect, the patch size of samples in HybridSN and 3DCNN is set to 25. The hyper-parameters of SSRN and SuperPCA are set as described in the original papers. We carry out all the methods for comparison, and all the experimental results in this paper are the averages of five result values obtained by independent runs for five times on RTX TITAN.

Three commonly used metrics are adopted in this paper to evaluate the classification results of HSIs obtained by different methods: Overall Accuracy (OA), Average Accuracy (AA), and Kappa coefficient (Kappa). OA represents the proportion of correctly classified samples to all samples. AA represents the average value of classification accuracy of each class. Kappa represents the degree of consistency between the classification result and the true labels. The larger the three metrics are, the better the classification performance is.

Suppose $n$ is the total number of samples, $m$ is the total number of classes, $N_i$ is the total number of samples of the actual class $i$, $N_i'$ is the total number of samples predicted as class $i$, and $C_{ii}$ represents the number of samples whose true class is $i$ and is predicted to be class $i$; then, OA can be expressed by Formula (8):

$$OA = \frac{1}{n} \sum_{i=1}^{m} C_{ii} \tag{8}$$

AA can be expressed by Formula (9):

$$AA = \frac{1}{m} \sum_{i=1}^{m} \frac{C_{ii}}{N_i} \tag{9}$$

Kappa can be expressed by Formula (10):

$$Kappa = \frac{P_o - P_e}{1 - P_e} \tag{10}$$

where $P_o = \frac{1}{n} \sum_{i=1}^{m} C_{ii}$, $P_e = \frac{1}{n^2} \sum_{i=1}^{m} N_i \times N_i'$.

### 3.3. The Results and Analyses of Experiments

#### 3.3.1. The Difference between Two-Branch Network and Single-Branch Network

In order to prove that the two-branch neural network plays an important role in feature extraction and feature fusion, we designed two single-branch networks (Net-X and Net-S) for comparison. Net-X means that only preprocessed HSI is used for feature extraction and classification, and Net-S means that only preprocessed MERSI is used for feature extraction and classification, whereas TBN-MERS performs feature extraction, feature fusion, and classification using both the preprocessed HSI and MERSI. The hyper-parameter settings of Net-X and Net-S are the same as those of TBN-MERS. The experimental results are shown in Table 6.

**Table 6.** Classification results of Net-X, Net-S, and TBN-MERS.

| Networks | Metrics | IP | PU | SA | HU |
|----------|---------|------|------|------|------|
| Net-X | OA (%) | 81.02 | 94.63 | 85.31 | 94.68 |
| | AA (%) | 87.70 | 94.33 | 91.74 | 95.39 |
| | Kappa (×100) | 78.44 | 92.88 | 83.59 | 94.25 |
| Net-S | OA (%) | 97.59 | 95.90 | 97.89 | 90.58 |
| | AA (%) | 98.97 | 96.37 | 98.65 | 91.90 |
| | Kappa (×100) | 97.23 | 94.58 | 97.66 | 89.83 |
| TBN-MERS | OA (%) | **98.13** | **99.74** | **99.35** | **97.51** |
| | AA (%) | **99.01** | **99.70** | **99.31** | **97.88** |
| | Kappa (×100) | **97.85** | **99.66** | **99.28** | **97.31** |

From Table 6, we can see that the contribution of HSI and MERSI to the final classification results is quite different on different datasets. On IP and SA datasets, Net-S performs much better than Net-X, indicating that MERSI contains more useful information of these two datasets for classification. As for PU and HU datasets, Net-X is a little inferior to Net-S on PU dataset but is much superior to Net-S on HU dataset, indicating that the features extracted from HSIs are also important for the final classification accuracy. Compared to Net-X and Net-S, TBN-MERS obtains the best classification accuracy on all the four datasets, showing the necessity of the feature fusion. TBN-MERS obtains more advanced features and complementary information by feature fusion, which thus improves the final classification results greatly. In a word, the proposed two-branch network TBN-MERS is effective in fusing the features of both the preprocessed HSI and MERSI, which is thus more efficient and robust in classification.

### 3.3.2. The Influence of the Number of Superpixels and the Patch Size

There are two important parameters in the proposed TBN-MERS: the number of superpixels $n_c$ and patch size $p$. $n_c$ is a hyper-parameter in ERS, which determines how many superpixels ERS divides the image into. If $n_c$ is large, it means that TBN-MERS performs a fine segmentation on the image and the size of each superpixel is small, obtaining detailed information of the image. On the contrary, if $n_c$ is small, it means that the size of each superpixel is big, and TBN-MERS pays more attention to the large-scale features, such as the consistency in a wide area. The patch size $p$ is the size of patch-based neighborhoods of the input samples. In brief, $n_c$ determines the richness of superpixel-scale spatial information, and $p$ determines the richness of neighborhood-scale spatial information. We conduct experiments on different combinations of $n_c$ and $p$ and plot the results in Figure 5, where different colors indicate the results corresponding to different $p$.

From Figure 5, it can be seen that TBN-MERS with a small $n_c$ and a small $p$ achieves better OA results on IP. Such results are reasonable. It can be observed from Figure 4a that IP has good consistency in large areas. A smaller $n_c$ means that the features are extracted from the superpixels with a larger size, and the pixels in the same superpixel have better consistency, whereas a larger $n_c$ makes TBN-MERS extract the features from the smaller superpixels, which will destroy the internal homogeneousness of the area. On the other hand, superpixel-scale information has a more important effect on the classification of IP, as shown in Section 3.3.1, where Net-S performs much better than Net-X on IP. Therefore, if a larger $p$ is used, it may introduce redundant information on the neighborhood scale and will interfere in the feature extraction and keep the model from extracting the useful features well. In short, a small $n_c$ and a small $p$ are good choices for IP. For PU, the reverse is true. TBN-MERS with a larger $n_c$ can achieve better classification results. When $n_c$ is large, a larger $p$ can further achieve better classification results. It can be observed from Figure 4b that the scene of PU is relatively complex, and the distribution of the ground objects is irregular, so a larger $n_c$ (corresponding to a smaller size of superpixel) enables the model to extract fine-grained superpixel-scale information, and a larger $p$ enables the model to extract richer neighborhood-scale information. For SA, which is homogeneous in

large areas, a smaller $n_c$ and a suitable $p$ achieve better classification results, which shows that the superpixel-scale information is more important for the classification of SA. For HU, the classification effect is not sensitive to the parameter changes, and many suitable $n_c$ and $p$ can lead to good classification results.

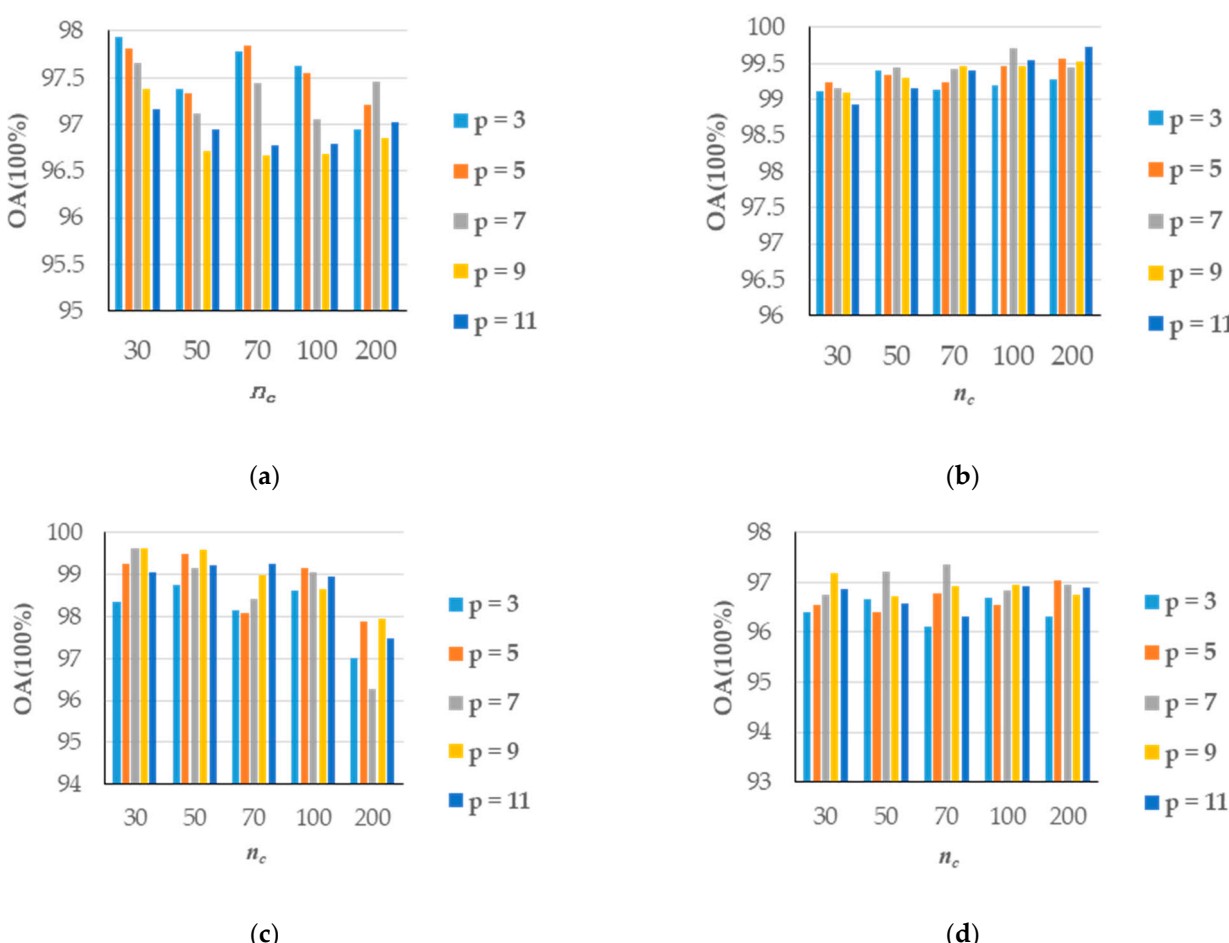

**Figure 5.** OA obtained on four datasets with different number of superpixels and patch size (**a**) IP, (**b**) PU, (**c**) SA, and (**d**) HU.

### 3.3.3. The Difference of Multi-Spectral Methods Based on Two Kinds of Superpixel Segmentation Methods, ERS and SLIC

TBN-MERS introduces the superpixel-scale spatial information in each band using the results of ERS segmentation as the priori information, which significantly improves the final classification results. How much does the final classification results depend on the effect of the superpixel segmentation? How will the proposed model of TBN-MERS perform if ERS is replaced by another superpixel segmentation method? We will investigate these issues in this section. As introduced in Section 2.1, there are many superpixel segmentation methods, among which both SLIC and ERS have excellent segmentation effect. We design a variant of TBN-MERS by replacing ERS with SLIC in the model of TBN-MERS, which is named as TBN-MSLIC. The comparison results between TBN-MSLIC and TBN-MERS are shown in Table 7. From Table 7, it can be seen that both TBN-MSLIC and TBN-MERS can achieve good performance with higher classification accuracy than that of Net-X in Section 3.3.1, indicating that the model of TBN-MERS is effective and can obtain higher classification accuracy than the single-branch network, which only uses the spectral information and the spatial information at the scale of patch-based neighborhood, even if ERS in TBN-MERS is replaced by other superpixel segmentation method. In addition, TBN-MERS performs better than TBN-MSLIC, especially on IP and SA datasets with large homogeneous regions.

According to the analyses shown in Section 3.3.1, the superpixel segmentation image contains more useful information of IP and SA datasets for classification. Therefore, the model using the better superpixel segmentation method will achieve better classification performance on IP and SA datasets. Figure 6 shows the results of superpixel segmentation obtained by SLIC and ERS on the 103rd band of PU set, respectively. The number of superpixels in Figure 6a,b is both 200. From Figure 6, it can be seen that both SLIC and ERS can extract the superpixel-scale spatial information. For example, both SLIC and ERS can segment the contour of the parking area at bottom left. However, there are also some differences. The boundary obtained by SLIC is smoother, without too many local details, whereas the boundary obtained by ERS is finer, and both the overall outlines and local details are well-described. In summary, the model of TBN-MERS is effective, where ERS can obtain better superpixel segmentation results than SLIC, enabling TBN-MERS to achieve good classification results on different datasets.

**Table 7.** The comparison results of TBN-MSLIC and TBN-MERS.

| Methods | Metrics | IP | PU | SA | HU |
|---|---|---|---|---|---|
| | OA (%) | 95.01 | 99.20 | 95.16 | 96.59 |
| TBN-MSLIC | AA (%) | 97.48 | 99.55 | 96.55 | 97.13 |
| | Kappa (×100) | 94.28 | 98.94 | 94.62 | 96.31 |
| | OA (%) | **98.13** | **99.74** | **99.35** | **97.51** |
| TBN-MERS | AA (%) | **99.01** | **99.70** | **99.31** | **97.88** |
| | Kappa (×100) | **97.85** | **99.66** | **99.28** | **97.31** |

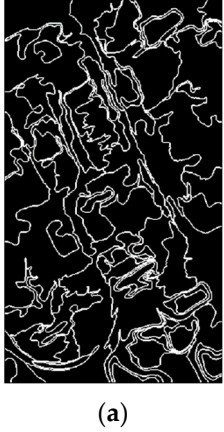 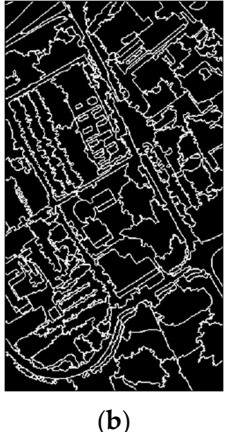

(**a**)  (**b**)

**Figure 6.** The segmentation image obtained by SLIC and ERS on the 103rd band of PU (**a**) SLIC (**b**) ERS.

3.3.4. The Different Effects of Segmentation Images Obtained by Applying ERS to Images on Multiple Bands and the First Principal Component

Some existing methods [27,28] used to extract the spectral–spatial information within each superpixel for classification by performing ERS on the first principal component of HSIs. However, different bands contain not only different spectral information but also different spatial information; the spectral–spatial information can hardly be sufficiently utilized by only choosing the first principal component. Figure 7 shows the results segmented by ERS on the first principal component and some bands of PU. Since each region in the image has different radar reflectivity on different bands, it can be seen from Figure 7 that the superpixel segmentation results obtained by ERS on different bands have significant differences. Comparing with the true label image of PU, the result of the first principal component seems to have good effect at first glance, but a different conclusion could be drawn after a careful comparison. For instance, for the parking area at the bottom left of the image (marked by a big rectangle in Figure 7a,c), the segmentation result of the 51st

band is better. For the circular building areas (marked by a small rectangle in Figure 7a,d,e), the 77th and 103rd bands have a better segmentation effect. It can also be noticed that the contours of each area on most bands are roughly similar, but there are differences in local details. This exactly reflects the difference between different bands. The segmentation result of the first principal component can obtain the segmentation result of the overall contour, which contains the mainly superpixel-scale spatial information. The obtained superpixels of different bands are overlapped and interlaced, which not only contain a wealth of superpixel-scale spatial information, but also include very fine local spatial information of different regions in different bands.

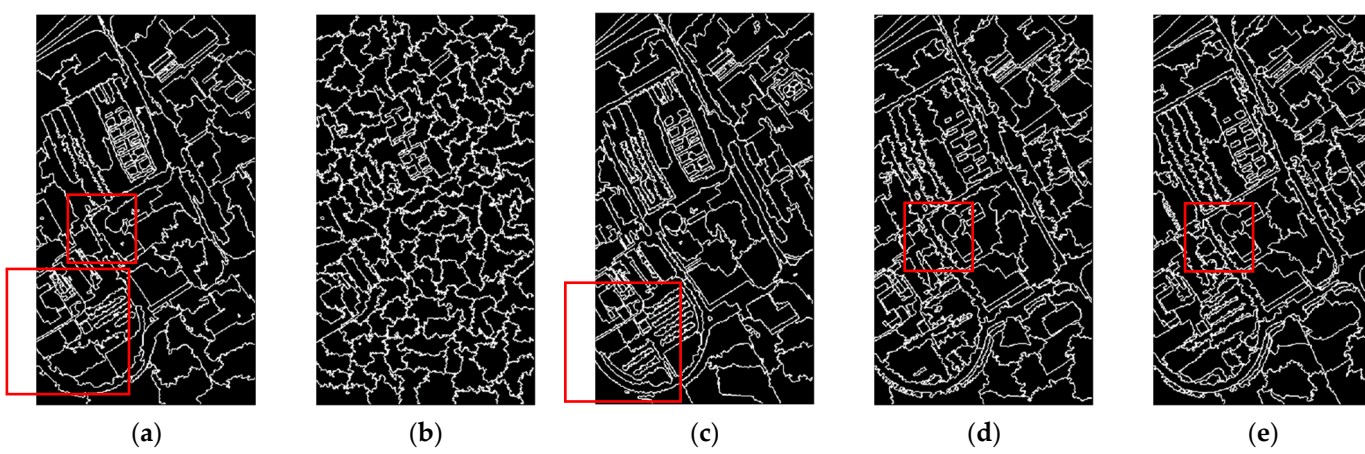

**Figure 7.** The segmentation results of ERS on different bands of PU: (**a**) the first principal component, (**b**) the 1st band, (**c**) the 51st band, (**d**) the 77th band, and (**e**) the 103rd band.

In order to prove that the segmentation image obtained by multi-spectral ERS contributes to better classification results, we designed another variant of TBN-MERS, which is denoted by PC-TBN-MERS. The details of PC-TBN-MERS are as follows. The dimension of preprocessed HSI is reduced by principal component analysis, and the first principal component is retained. ERS is applied on the first principal component to obtain a superpixel segmentation result. The superpixel segmentation result is copied and stacked to form a data cube until it has the same size as that of the HSI. We take the obtained data cube and the preprocessed HSI as the input of two-branch neural network. Therefore, the only difference between PC-TBN-MERS and TBN-MERS lies in that the former only performs ERS on the first principal component (noted as FPC-ERS), and the latter performs ERS on all the bands (i.e., multi-spectral ERS). The comparison results are shown in Table 8.

**Table 8.** Classification results of PC-TBN-MERS and TBN-MERS.

| Methods | Metrics | IP | PU | SA | HU |
|---|---|---|---|---|---|
| | OA (%) | 86.58 | 97.80 | 93.79 | 96.28 |
| PC-TBN-MERS | AA (%) | 91.91 | 97.78 | 95.40 | 96.68 |
| | Kappa (×100) | 84.72 | 97.08 | 93.10 | 95.98 |
| | OA (%) | **98.13** | **99.74** | **99.35** | **97.51** |
| TBN-MERS | AA (%) | **99.01** | **99.70** | **99.31** | **97.88** |
| | Kappa (×100) | **97.85** | **99.66** | **99.28** | **97.31** |

From the metrics in Table 8, it can be seen that TBN-MERS has achieved much better performance than PC-TBN-MERS on IP and SA datasets, demonstrating that multi-spectral ERS is necessary. In addition, as shown in Figure 4, IP and SA datasets have several large homogeneous regions. This result also indicates that FPC-ERS is able to extract details of the image, but it is not good at finding the homogeneousness in a large area. However,

multi-spectral ERS is good at both aspects, making TBN-MERS obtain better classification results on all kinds of tested datasets.

### 3.3.5. The Effect of Using Different Fusion Methods in TBN-MERS

From the experimental results in Section 3.3.1, it can be seen that the proposed two-branch network TBN-MERS is effective in fusing the features of both the preprocessed HSI and MERSI, where the data fusion is performed after the flatten layer. If the data fusion is performed at earlier stages, could the final classification accuracy be higher or not? We will investigate this issue in this section.

We designed two variants of TBN-MERS. In the feature extraction process, the features of the same depth layer from different branches are fused through skip connections, including the introduction of MERSI features into the HIS feature extraction process, marked as TBN(M2H)-MERS, whose network structure is shown in Figure 8a, and the introduction of HIS features into MERSI feature extraction process, marked as TBN(H2M)-MERS, whose network structure is shown in Figure 8b. Table 9 lists the corresponding experimental results. From Table 9, it can be found that neither TBN(M2H)-MERS nor TBN(H2M)-MERS achieves better classification results than TBN-MERS.

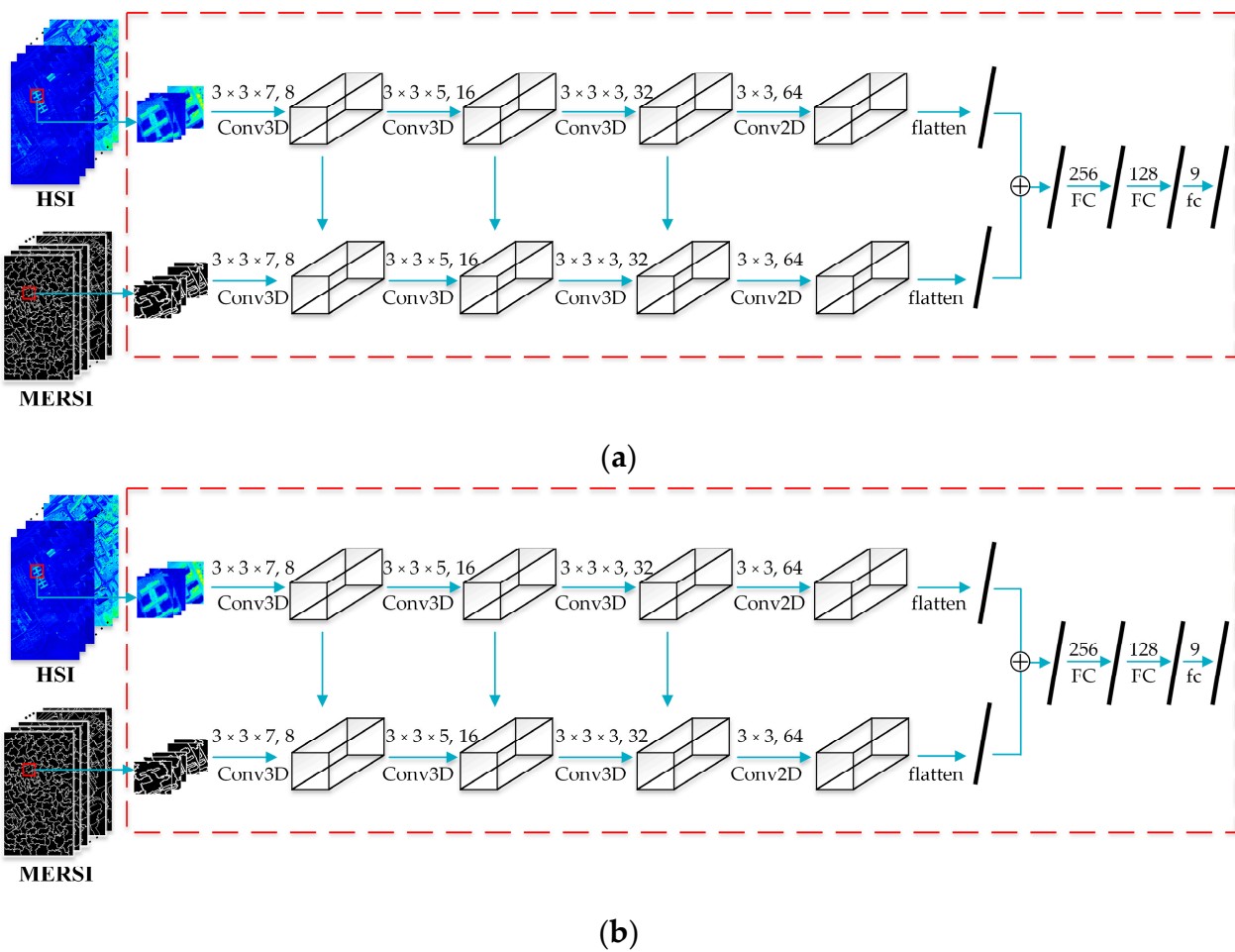

**Figure 8.** Network structures of two variants of TBN-MERS: (**a**) TBN(M2H)-MERS and (**b**) TBN(H2M)-MERS.

**Table 9.** Classification results of TBN(M2H)-MERS, TBN(H2M)-MERS, and TBN-MERS.

| Methods | Metrics | IP | PU | SA | HU |
|---|---|---|---|---|---|
| TBN(M2H)-MERS | OA (%) | 97.74 | 97.74 | 99.12 | 94.81 |
| | AA (%) | 98.99 | 98.29 | 98.94 | 95.72 |
| | Kappa ($\times 100$) | 97.41 | 97.01 | 99.02 | 94.39 |
| TBN(H2M)-MERS | OA (%) | 93.62 | 98.94 | 97.34 | 96.34 |
| | AA (%) | 96.41 | 99.30 | 97.14 | 96.87 |
| | Kappa ($\times 100$) | 92.70 | 98.60 | 97.04 | 96.05 |
| TBN-MERS | OA (%) | **98.13** | **99.74** | **99.35** | **97.51** |
| | AA (%) | **99.01** | **99.70** | **99.31** | **97.88** |
| | Kappa ($\times 100$) | **97.85** | **99.66** | **99.28** | **97.31** |

Why does information fusion at earlier stages not lead to better classification results? In our opinion, the reason is as follows. The data of HSIs contain the values of the electromagnetic reflection intensity of ground objects, and the data of MERSI contain the label values obtained by superpixel segmentation. The data distribution of HSI and MERSI is quite different. Therefore, if the feature fusion is carried out at earlier stages of the network, it will cause interference, which is not conducive to learning key features. In contrast, after the flatten layer, the network has extracted the advanced features of HSI and MERSI, and the features at this time represent the essential information, where the feature fusion will achieve better classification results. Therefore, we choose the structure of TBN-MERS to implement the information fusion, which is more concise and more effective.

3.3.6. Comparison between TBN-MERS and Other Methods

We compare TBN-MERS with five existing methods, including SVM [16], 3DCNN [24], SSRN [25], HybridSN [26], and SuperPCA [28]. SVM only takes the spectral features as input and can achieve satisfactory classification results even with limited samples. It has the advantages of good robustness and good generalization, so it is often used as a benchmark for classification of HSIs. Archibald discussed the application of SVM in hyperspectral image classification in 2007 [16]. We refer to the settings in that paper [16] and train the SVM classifier based on radial basis function to classify HSIs according to the spectral features of HSIs. SSRN was proposed by Zhong et al. in 2017 [25]. SSRN combined multiple 3D convolutional layers into spatial feature extraction modules and spectral feature extraction modules according to the residual structure to extract spectral–spatial features. SSRN connected these modules in series to perform end-to-end feature extraction and classification. In 2018, Hamida et al. [24] studied the classification effect of 3D convolutional networks with different numbers and different structures on HSIs. We choose the 3DCNN network structure consisting of four layers of 3D convolution and a fully connected layer proposed by Hamida for comparison. HybridSN was proposed by Roy et al. in 2019 [26]. It firstly used multiple 3D convolutional layers to extract the joint spectral–spatial features of HSIs, then used 2D convolution to extract the spatial information of the feature map, and finally, adopted a multi-layer fully connected layer to perform classification. It not only ensured the feature extraction ability of the model, but also reduced the model parameters. Jiang et al. proposed SuperPCA in 2018 [28]. In SuperPCA, the first principal component of HSI was segmented by multi-scale ERS, and then the principal component analysis (PCA) was applied to reduce the dimension of HSIs within the obtained superpixels. Subsequently, SuperPCA trained classifiers on the reduced data at each scale and obtained the final classification results through decision fusion.

The results and analyses of six different methods on four datasets are as follows. In the following tables, the best values are marked in bold, and the second best values are marked with underlines for the convenience of the readers. The numbers in parentheses are the standard deviation.

(1) Results and analyses of IP: Table 10 shows the OA, AA, and Kappa of the six different methods on IP. From Table 10, it can be seen that TBN-MERS achieves an overall

classification accuracy of 98.13%, which is increased by 26.27%, 15.97%, 9.75%, 4.36%, and 3.07% over SVM, 3DCNN, SSRN, HybridSN, and SuperPCA, respectively. Except that the classification accuracy of TBN-MERS in class 3 is slightly lower than that of HybridSN, TBN-MERS has achieved the highest classification accuracy in other classes. Both TBN-MERS and SuperPCA use ERS for feature extraction, and TBN-MERS and HybridSN use a similar network structure for feature extraction. Yet, TBN-MERS has achieved higher classification accuracy, showing that the use of multi-spectral ERS and the structure of the two-branch neural network both contribute to the final results.

**Table 10.** Classification results of different methods on IP with 50 samples per class.

| Class Number | SVM [16] | 3DCNN [24] | SSRN [25] | HybridSN [26] | SuperPCA [28] | TBN-MERS |
|---|---|---|---|---|---|---|
| 1 | 66.11 | 99.37 | 0.00 | **100.00** | **100.00** | **100.00** |
| 2 | 63.91 | 78.03 | 76.71 | 87.80 | 93.76 | **98.21** |
| 3 | 64.17 | 70.64 | 86.27 | **96.76** | 89.10 | 94.64 |
| 4 | 84.38 | 61.16 | **100.00** | 99.35 | 95.19 | **100.00** |
| 5 | 90.71 | 88.95 | 94.67 | 97.50 | 97.00 | **98.33** |
| 6 | 92.79 | 93.52 | 97.41 | 96.91 | 95.29 | **99.94** |
| 7 | 85.55 | 98.88 | 0.00 | **100.00** | 85.71 | **100.00** |
| 8 | 95.23 | 97.60 | **100.00** | **100.00** | 99.53 | **100.00** |
| 9 | 88.00 | 94.84 | 0.00 | **100.00** | 90.00 | **100.00** |
| 10 | 72.14 | 64.13 | 88.21 | 93.55 | 81.13 | **97.22** |
| 11 | 57.04 | 90.11 | 87.31 | 89.47 | 85.07 | **96.92** |
| 12 | 70.01 | 73.23 | 74.11 | 94.91 | 93.74 | **98.96** |
| 13 | 98.58 | 91.95 | 99.31 | **100.00** | 99.35 | **100.00** |
| 14 | 83.09 | 94.71 | 98.01 | 97.99 | 97.20 | **100.00** |
| 15 | 70.47 | 77.04 | 99.94 | 97.91 | 98.81 | **100.00** |
| 16 | 97.67 | 77.40 | 99.05 | **100.00** | 97.83 | **100.00** |
| OA (%) | 71.86 (0.34) | 82.16 (1.47) | 88.38 (1.58) | 93.77 (1.22) | 95.06 (1.24) | **98.13** (0.21) |
| AA (%) | 79.99 (0.93) | 84.47 (1.58) | 75.06 (1.07) | 97.01 (0.55) | 96.70 (1.00) | **99.01** (0.09) |
| Kappa (×100) | 68.22 (0.41) | 79.72 (1.65) | 86.73 (1.79) | 92.88 (1.38) | 94.32 (1.42) | **97.85** (0.24) |

The samples in different classes in IP are seriously unbalanced. For instance, the samples of class 1, class 7, and class 9 are very few, the number of which is 46, 28, and 20, respectively. When the number of training set samples is limited, the imbalance of the samples in different categories may make it difficult for the model to learn the features of the classes with limited samples. As we can see in Table 10 for SSRN, the classification accuracy values of these three classes are zero. In class 1, 3DCNN, HybridSN, SuperPCA and TBN-MERS achieve good classification results. In class 7 and class 9, only HybridSN and TBN-MERS achieve good classification results with the value of 100%. These results demonstrate that TBN-MERS is able to learn the features of classes with limited samples effectively, indicating it has the characteristic of good generalization and robustness.

Figure 9 shows the predicted label images of the six different methods on IP. From Figure 9, we can see that there is a certain amount of salt and pepper noise in the results of SVM and 3DCNN, whereas the results of HybridSN, SSRN, SuperPCA, and TBN-MERS are relatively close to the true label image. On the whole, the visual effect of the classification obtained by TBN-MERS is the best among all the compared methods.

(2) Results and analyses of PU: Table 11 shows the OA, AA, and Kappa of six different methods on PU. From Table 11, it can be seen that TBN-MERS achieves an overall classification accuracy of 99.86%, which is increased by 17.13%, 11.48%, 4.78%, 4.32%, and 6.62% over SVM, 3DCNN, SSRN, HybridSN, and SuperPCA, respectively. In each class, TBN-MERS has achieved the highest classification accuracy. Both HybirdSN and SSRN directly use hyperspectral data as input. SuperPCA uses the superpixel segmentation results of the first principal component for feature extraction. Compared with these meth-

ods, TBN-MERS has achieved a significant improvement in classification accuracy. For PU, the features extracted by the multi-spectral ERS and the combined spectral–spatial information extracted by TBN-MERS play an important role in improving the classification accuracy. Figure 10 shows the predicted label images of the six different methods. From Figure 10, we can see that SVM, 3DCNN, and SuperPCA have more misclassifications, whereas HybridSN, SSRN, and TBN-MERS results are very close to the true label image. On the whole, the visual effect of the classification obtained by TBN-MERS is the most satisfactory among all the compared methods.

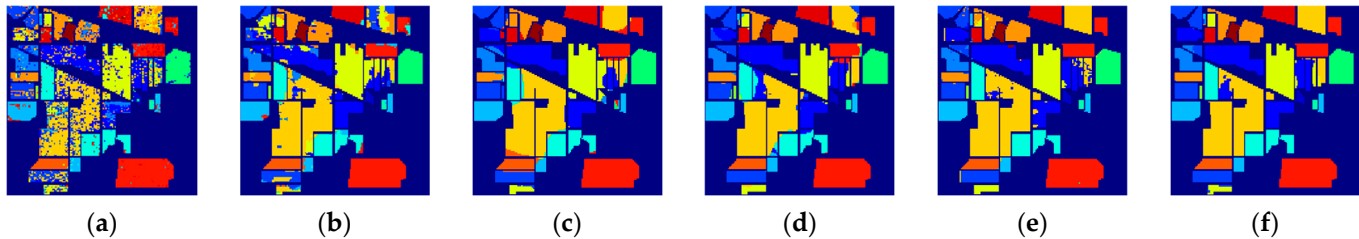

| (a) | (b) | (c) | (d) | (e) | (f) |

**Figure 9.** The predicted label images of different methods on IP: (**a**) SVM, (**b**) 3DCNN, (**c**) SSRN, (**d**) HybridSN, (**e**) SuperPCA, and (**f**) TBN-MERS.

**Table 11.** Classification results of different methods on PU with 50 samples per class.

| Class Number | SVM [16] | 3DCNN [24] | SSRN [25] | HybridSN [26] | SuperPCA [28] | TBN-MERS |
|---|---|---|---|---|---|---|
| 1 | 77.34 | <u>95.27</u> | 92.55 | 87.33 | 70.66 | **99.74** |
| 2 | 80.62 | 94.78 | <u>98.40</u> | 98.38 | 80.93 | **99.89** |
| 3 | 79.59 | 66.67 | <u>97.62</u> | 94.74 | 93.70 | **99.50** |
| 4 | <u>95.08</u> | 93.30 | 70.38 | 94.10 | 80.23 | **98.95** |
| 5 | 99.32 | 99.42 | 99.67 | <u>99.84</u> | 96.60 | **100.00** |
| 6 | 79.97 | 75.00 | 99.75 | <u>99.86</u> | 87.73 | **100.00** |
| 7 | 93.06 | 63.18 | 97.51 | <u>99.93</u> | 92.97 | **100.00** |
| 8 | 84.86 | 84.21 | <u>99.38</u> | 89.00 | 91.88 | **99.53** |
| 9 | <u>99.86</u> | 96.08 | 68.34 | 93.73 | **100.00** | 100.00 |
| OA (%) | 82.73 (1.78) | 88.38 (0.99) | 95.08 (1.27) | <u>95.54</u> (0.71) | 93.24 (0.67) | **99.86** (0.07) |
| AA (%) | 87.75 (0.42) | 85.32 (0.94) | 91.51 (1.00) | <u>95.21</u> (0.70) | 94.42 (0.37) | **99.77** (0.08) |
| Kappa (×100) | 77.78 (2.08) | 84.69 (1.25) | 93.47 (1.66) | <u>94.09</u> (0.94) | 91.10 (0.85) | **99.64** (0.09) |

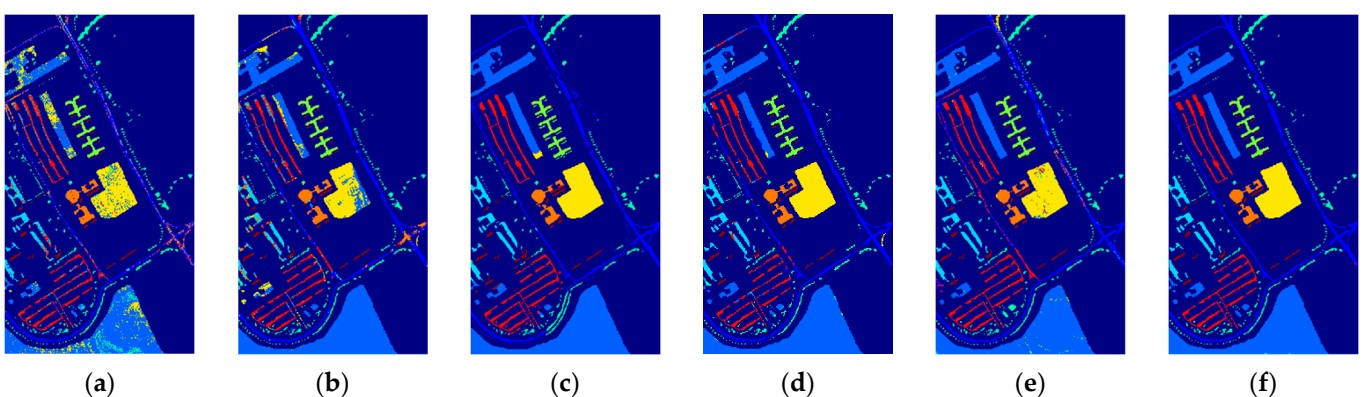

| (a) | (b) | (c) | (d) | (e) | (f) |

**Figure 10.** The predicted label images of different methods on PU: (**a**) SVM, (**b**) 3DCNN, (**c**) SSRN, (**d**) HybridSN, (**e**) SuperPCA, and (**f**) TBN-MERS.

(3) Results and analyses of SA: Table 12 shows the OA, AA, and Kappa of six different methods on SA. In particular, due to the relatively simple scenarios of SA and the continuous distribution of ground objects, the classification of SA is easier to a degree compared with other datasets. To fully explore the classification performance of different methods, in the phase of training, only five training samples are randomly selected for each class. From Table 12, it can be seen that TBN-MERS achieves an overall classification accuracy of 99.31%, which is increased by 28.79%, 16.84%, 18.46%, 8.49%, and 25.67% over SVM, 3DCNN, SSRN, HybridSN, and SuperPCA, respectively. As considering a certain class, TBN-MERS has not achieved the highest classification accuracy in many classes. The accuracy of TBN-MERS is lower than 3DCNN in class 5, class 6, and class 16. It is lower than SSRN in class 9. It is lower than HybridSN in class 14 and class 16. It is lower than SuperPCA in class 15. However, the classification accuracy gap is very small in these classes, whereas the accuracy of TBN-MERS far exceeds other methods in other classes, such as class 8 and class 10. This shows that TBN-MERS has stronger generalization ability and better robustness for different types of ground objects on SA. Compared with other datasets, SA has the characteristics of concentrated ground object distribution, relatively regular shape of ground objects, high ground object consistency, and high consistency within the sample class, so it is especially suitable for multi-spectral ERS. The obtained superpixels well mark the characteristics of the distribution of ground objects. Even if the samples are limited and the neighborhood-scale spatial information is limited, TBN-MERS can also classify the test samples well by fully extracting the superpixel-scale spatial information of SA and filtering out the most representative features for classification. Figure 11 shows the predicted label images of the six different methods. From Figure 11, we can see that the predicted label images of SVM, 3DCNN, SSRN, HybridSN, and SuperPCA all have many pixels that are wrongly classified, whereas the result of TBN-MERS is almost the same as the true label image. On the whole, the visual effect of the classification obtained by TBN-MERS is the best among all the compared methods.

**Table 12.** Classification results of different methods on SA with 5 samples per class.

| Class Number | SVM [16] | 3DCNN [24] | SSRN [25] | HybridSN [26] | SuperPCA [28] | TBN-MERS |
|---|---|---|---|---|---|---|
| 1 | 98.35 | 73.44 | 99.67 | 98.65 | **100.00** | **100.00** |
| 2 | 81.50 | 97.13 | 95.17 | 97.32 | 73.53 | **100.00** |
| 3 | 37.44 | 99.48 | 96.14 | 97.54 | 87.27 | **100.00** |
| 4 | 97.99 | 98.52 | 93.80 | 83.94 | 70.27 | **99.98** |
| 5 | 93.92 | **99.48** | 80.07 | 91.73 | 51.63 | 97.23 |
| 6 | 97.20 | **99.96** | 99.09 | 97.92 | 85.31 | 99.39 |
| 7 | 99.02 | 91.03 | 92.90 | 99.25 | 72.97 | **100.00** |
| 8 | 37.31 | 66.54 | 71.32 | 85.45 | 33.37 | **98.95** |
| 9 | 97.97 | 86.99 | **100.00** | 94.20 | 53.34 | 99.93 |
| 10 | 24.32 | 86.16 | 82.38 | 91.03 | 66.27 | **98.75** |
| 11 | 83.55 | 95.99 | 73.93 | 94.93 | 96.33 | **99.34** |
| 12 | 96.13 | 93.55 | 62.59 | 97.00 | 78.15 | **99.76** |
| 13 | 98.77 | 89.27 | 49.98 | 96.11 | 95.28 | **99.62** |
| 14 | 88.45 | 85.89 | 94.70 | **99.26** | 71.74 | 98.74 |
| 15 | 57.51 | 59.00 | 52.28 | 75.95 | **100.00** | 99.59 |
| 16 | 71.67 | **98.78** | 77.88 | 97.67 | 74.97 | 97.03 |
| OA (%) | 70.52 (1.19) | 82.47 (0.83) | 80.85 (4.37) | 90.82 (1.89) | 73.64 (3.53) | **99.31** (0.13) |
| AA (%) | 78.82 (0.81) | 88.83 (1.01) | 82.62 (5.78) | 93.62 (0.89) | 81.35 (2.46) | **99.27** (0.13) |
| Kappa (×100) | 67.37 (1.29) | 80.39 (0.92) | 78.71 (4.88) | 89.78 (2.12) | 70.92 (3.65) | **99.23** (0.15) |

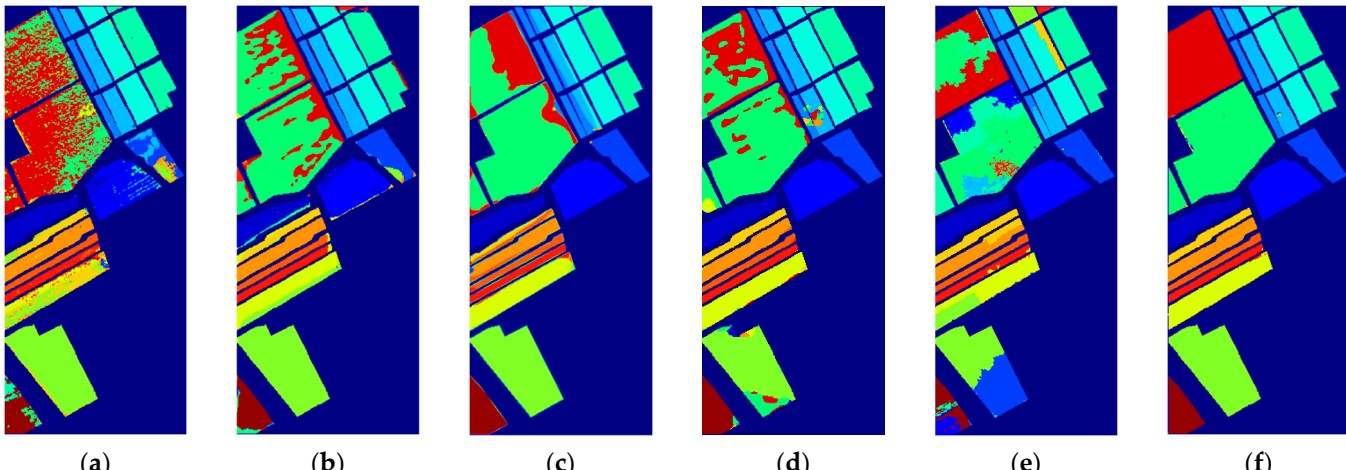

**Figure 11.** The predicted label images of different methods on SA: (**a**) SVM, (**b**) 3DCNN, (**c**) SSRN, (**d**) HybridSN, (**e**) Super-PCA, and (**f**) TBN-MERS.

(4) Results and analyses of HU: Table 13 shows the OA, AA, and Kappa of the six different methods on HU. HU scene is relatively complex, the ground objects are small, and the distribution is not continuous, which certainly introduce some difficulty for the classification. From Table 13, it can be seen that TBN-MERS achieves an overall classification accuracy of 97.52%, which is increased by 39.65%, 14.16%, 4.75%, 3.06%, and 5.68% over SVM, 3DCNN, SSRN, HybridSN, and SuperPCA, respectively. As considering a certain class, the accuracy of TBN-MERS is lower than HybridSN in class 5 and class 6 and lower than SSRN in class 8. TBN-MERS has achieved the highest classification accuracy in other classes. For HU, SSRN and HybridSN have achieved relatively high classification accuracy, indicating that the neighborhood-scale spatial information has an important effect on the classification results. SuperPCA also has achieved good classification accuracy, indicating that the superpixel-scale spatial information has an important effect on the classification accuracy. TBN-MERS has achieved the highest classification accuracy, indicating that the two-branch network fully combines the superpixel-scale and neighborhood-scale spatial information through feature fusion. Figure 12 shows the predicted label images of the six different methods. From Figure 12, we can see that some misclassified samples can be seen in the predicted label images of SVM and 3DCNN, whereas SSRN, HybridSN, SuperPCA, and TBN-MERS are relatively similar with the true label image. On the whole, the visual effect of the classification obtained by TBN-MERS is the most satisfactory among all the compared methods.

**Table 13.** Classification results of different methods on HU with 50 samples per class.

| Class Number | SVM [16] | 3DCNN [24] | SSRN [25] | HybridSN [26] | SuperPCA [28] | TBN-MERS |
|---|---|---|---|---|---|---|
| 1 | 84.82 | 86.91 | 86.71 | 89.34 | <u>91.51</u> | **95.25** |
| 2 | 84.30 | 79.08 | 87.39 | <u>93.32</u> | 83.31 | **99.13** |
| 3 | 98.91 | 90.60 | 97.74 | <u>99.97</u> | 99.71 | **100.00** |
| 4 | 90.46 | 82.42 | 84.58 | <u>94.67</u> | 92.29 | **98.29** |
| 5 | 86.66 | 88.86 | 99.75 | **99.89** | 97.15 | <u>99.83</u> |
| 6 | 84.29 | 66.38 | 94.26 | **100.00** | 87.89 | <u>97.92</u> |
| 7 | 21.67 | 87.21 | <u>88.68</u> | 88.11 | 82.92 | **96.33** |
| 8 | 19.88 | 77.21 | **91.22** | 85.37 | 77.55 | <u>90.78</u> |
| 9 | 82.07 | 85.39 | <u>88.58</u> | 87.65 | 81.95 | **92.74** |
| 10 | 3.21 | 82.99 | <u>99.17</u> | 99.08 | 91.59 | **100.00** |
| 11 | 56.07 | 83.76 | <u>98.50</u> | 97.49 | 90.95 | **99.75** |
| 12 | 2.16 | 75.74 | 93.88 | <u>98.03</u> | 78.36 | **98.41** |
| 13 | 10.27 | 91.38 | 94.90 | <u>99.12</u> | 72.97 | **99.95** |
| 14 | 96.61 | 76.71 | **100.00** | **100.00** | <u>96.85</u> | **100.00** |
| 15 | <u>99.04</u> | 94.37 | 98.05 | **100.00** | 98.92 | **100.00** |

**Table 13.** *Cont.*

| Class Number | SVM [16] | 3DCNN [24] | SSRN [25] | HybridSN [26] | SuperPCA [28] | TBN-MERS |
|---|---|---|---|---|---|---|
| OA (%) | 57.87 | 83.36 | 92.77 | 94.46 | 91.84 | **97.52** |
| | (0.56) | (2.27) | (0.77) | (0.52) | (1.69) | (0.25) |
| AA (%) | 61.36 | 83.27 | 93.56 | 95.47 | 92.24 | **97.89** |
| | (0.63) | (1.97) | (0.73) | (0.43) | (1.39) | (0.20) |
| Kappa | 54.74 | 82.04 | 92.18 | 94.00 | 91.18 | **97.32** |
| (×100) | (0.61) | (2.45) | (0.83) | (0.56) | (1.83) | (0.27) |

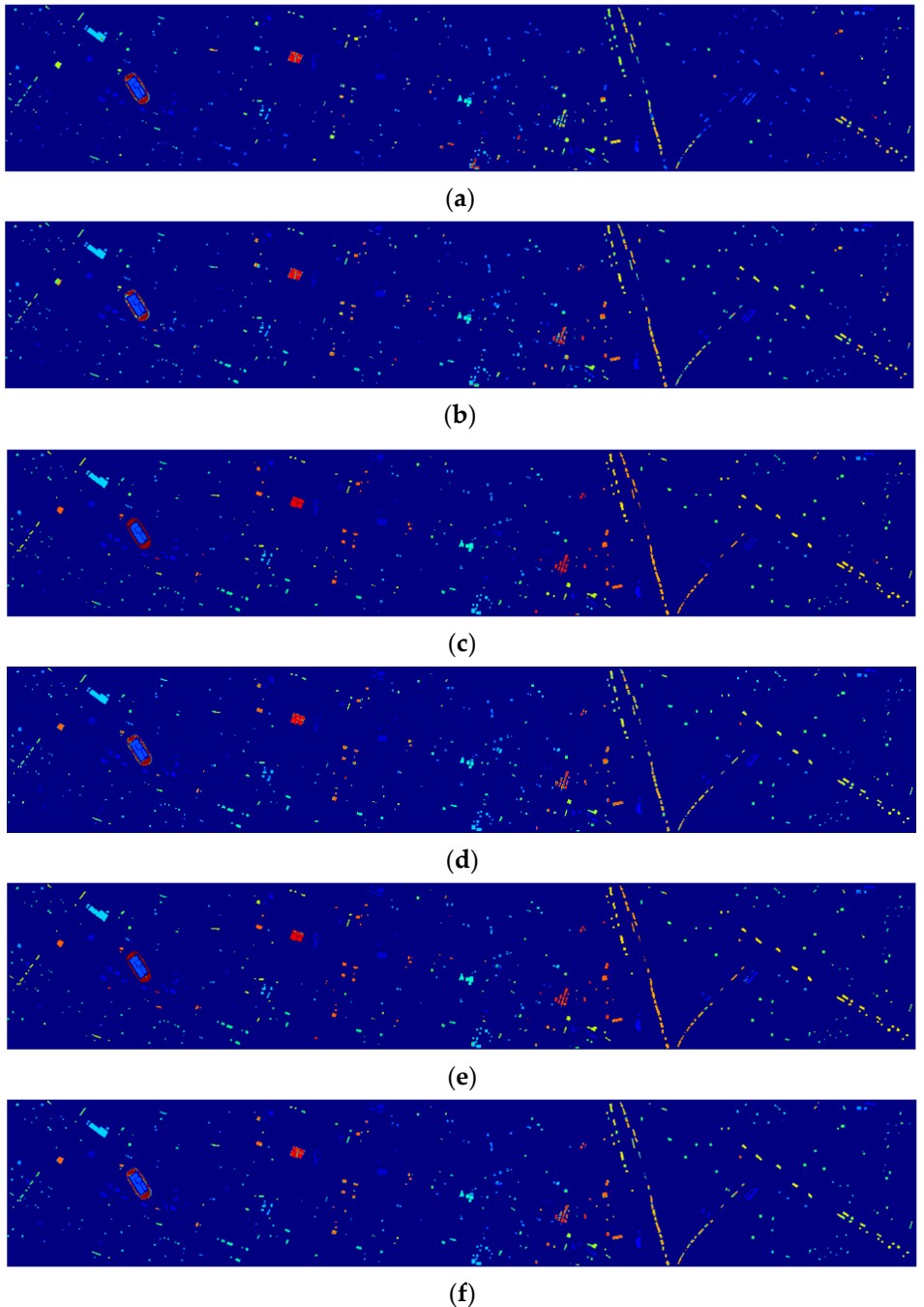

**Figure 12.** The predicted label images of different methods on HU: (**a**) SVM, (**b**) 3DCNN, (**c**) SSRN, (**d**) HybridSN, (**e**) SuperPCA, and (**f**) TBN-MERS.

(5) Comparison about the training loss of deep learning methods on different datasets: For the deep learning methods including 3DCNN, SSRN, HybridSN, and our TBN-MERS, we compare their convergence curves of training loss on different datasets, as shown in Figure 13. Figure 13a–d show the training loss curves of four deep learning methods on IP, PU, SA, and HU. In Figure 13, we can see that 3DCNN, HybridSN, and TBN-MERS converge faster, and their final loss is smaller than SSRN. The convergence of TBN-MERS in the training process is not the fastest, and there are certain fluctuations in the training process, but in the end, the highest classification accuracy can be achieved on the testing set, which illustrates that the two-branch structure and the feature fusion of superpixel-scale and neighborhood-scale make TBN-MERS more robust and have better generalization performance.

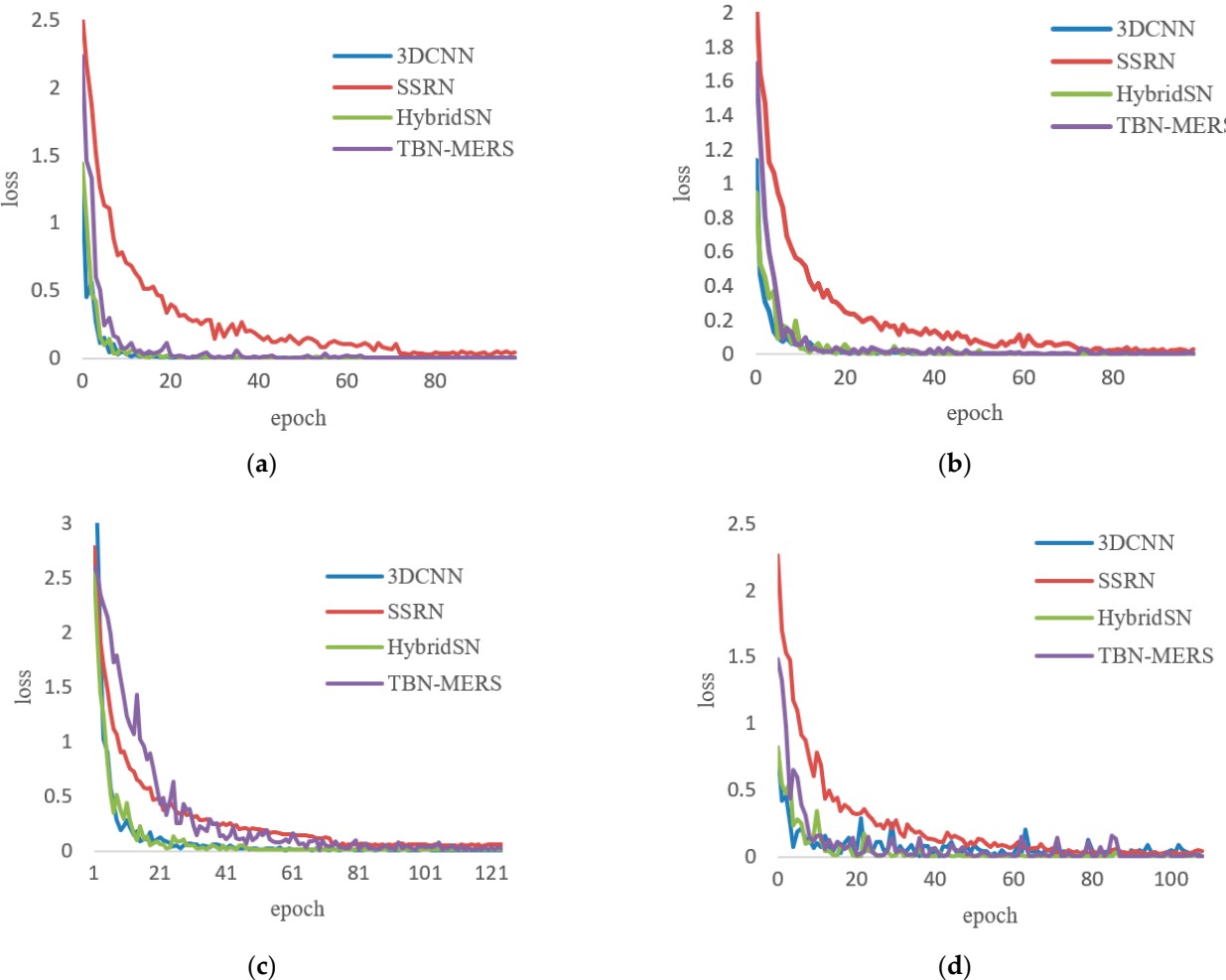

**Figure 13.** Training loss curves of four deep learning methods on different datasets: (**a**) IP, (**b**) PU, (**c**) SA, and (**d**) HU.

(6) Comparison about the training time of several deep learning methods: To compare the training time of several deep learning methods, further experiments are conducted, and the results are shown in Table 14. TBN-MERS needs to implement the superpixel segmentation first and then carry out the training stage, so we record the segmentation time, training time, and total time. From Table 14, in terms of training time, the training time of TBN-MERS is the smallest, because it uses a smaller patch size, making it require less computation. However, its segmentation time is relatively long, so the total time is not the minimal. At the same time, comparing the segmentation time of different datasets, we can find that the segmentation time is longer for datasets with larger size and more

channels. This cost is acceptable, as the final classification results are improved greatly, and the increasing of the total time is limited.

**Table 14.** Time for deep learning methods on different datasets.

| Methods | Operation | IP | PU | SA | HU |
|---------|-----------|------|------|------|------|
| 3DCNN | Training | 530 s | 121 s | 421 s | 299 s |
| SSRN | Training | 794 s | 257 s | 114 s | 566 s |
| HybridSN | Training | 335 s | 184 s | 50 s | 536 s |
| | Segmentation | 121 s | 152 s | 230 s | 646 s |
| TBN-MERS | Training | 70 s | 60 s | 38 s | 114 s |
| | Total time | 191 s | 212 s | 268 s | 760 s |

## 4. Discussion

It can be seen from the above results that the performance of different methods on different datasets is different, but TBN-MERS always achieves best classification accuracy with limited samples. The main reason why TBN-MERS can achieve highest classification accuracy is as follows. On the one hand, we apply ERS to each band of the preprocessed HSI. Each superpixel segmentation image obtained contains the spatial distribution of ground objects in the superpixel scale. The segmentation results of different bands are different, and the obtained superpixels on different bands are crossing and overlapping, which reflects the difference of the spectral information contained in different bands indirectly. That is to say, the superpixel segmentation images not only directly contain rich spatial information but also indirectly contain spectral information. In addition, in the superpixel segmentation images, the values of all pixels within the same superpixel are all equal to the serial number of the superpixel (i.e., the number in the range of 1 to $n_c$), which is essentially a filtering and denoising process for HSIs. Therefore, a single-branch network that uses only preprocessed segmentation images (i.e., Net-S in Section 3.3.1) can also achieve good classification accuracy. On the other hand, the proposed two-branch network not only utilizes superpixel-scale information in the HSI, but also extracts lots of detailed spectral–spatial information at the neighborhood-scale from the preprocessed HSI. The feature fusion designed in TBN-MERS further improves the classification accuracy and improves the robustness and generalization of TBN-MERS.

From the results of above experiments, we also find some limitations of TBN-MERS. Compared with several other deep learning methods, TBN-MERS mainly relies on the extra features provided by multi-spectral ERS to improve the classification accuracy. The time cost of multi-spectral ERS increases as the size of dataset becomes larger, which will affect the efficiency of TBN-MERS to a certain extent. Fortunately, such extra time cost is limited, as shown in Table 14, which is worthwhile, considering the significant improvement in classification results.

## 5. Conclusions

This paper proposed a two-branch convolutional neural network based on multi-spectral entropy rate superpixel segmentation (TBN-MERS) for hyperspectral image (HSI) classification. The multi-spectral entropy rate superpixel (ERS) segmentation, i.e., performing ERS segmentation on each band of the HSI, enables TBN-MERS to extract rich spectral–spatial information at the superpixel scale with the effect of filtering and denoising, which significantly improves the final classification accuracy. The feature fusion implemented by a two-branch network in TBN-MERS, which utilize both the superpixel-scale information and the spectral–spatial information at the neighborhood-scale in the HSI, improves the classification accuracy further, and improves the robustness and generalization of TBN-MERS.

On the whole, TBN-MERS can achieve good classification results on different kinds of datasets with limited training samples. For datasets with large homogeneous areas, TBN-MERS is able to extract such homogeneousness within the same regions effectively by using

the multi-spectral ERS segmentation to extract superpixel-scale information and obtains high classification accuracy finally. For datasets with fragmentary and dispersed ground objects, where the superpixel-scale information plays a less important role, TBN-MERS can fully mine the neighborhood-scale information and fuse it with the superpixel-scale information to ensure the high classification accuracy. In summary, both the multi-spectral ERS segmentation and the feature fusion contribute to the excellent classification results of TBN-MERS. TBN-MERS is designed with a concise and flexible frame, and its classification performance could be improved further if a better spectral–spatial feature extraction scheme is adopted in the first branch or a segmentation algorithm with performance better than ERS is applied in the second branch. In the future, we will try to design deep neural networks to implement unsupervised or semi-supervised image segmentation, which may help to improve the classification results further.

**Author Contributions:** Conceptualization, Z.D. and C.M.; methodology, Z.D. and C.M.; software, Z.D.; investigation, Z.D.; data curation, Y.L.; writing—original draft preparation, Z.D.; writing—review and editing, C.M. and Y.L.; visualization, Z.D. and Y.L.; supervision, C.M.; funding acquisition, C.M. and Y.L. All authors have read and agreed to the published version of the manuscript.

**Funding:** This research was funded by the National Natural Science Foundation of China, grant number 62077038, 61672405, 61871306, 62176196, 61836009 and 61871310.

**Institutional Review Board Statement:** Not applicable.

**Informed Consent Statement:** Not applicable.

**Data Availability Statement:** The data used in this study are available at http://www.ehu.eus/ccwintco/index.php/Hyperspectral_Remote_Sensing_Scenes#Indian_Pines (accessed on 16 February 2022) (Indian Pines), http://www.ehu.eus/ccwintco/index.php/Hyperspectral_Remote_Sensing_Scenes#Pavia_University_scene (accessed on 16 February 2022) (Pavia University), http://www.ehu.eus/ccwintco/index.php/Hyperspectral_Remote_Sensing_Scenes#Salinas_scene (accessed on 16 February 2022) (Salinas), and https://hyperspectral.ee.uh.edu/?page_id=459 (accessed on 16 February 2022) (Houston).

**Acknowledgments:** The authors would like to thank the Hyperspectral Image Analysis group and the NSF Funded Center for Airborne Laser Mapping (NCALM) at the University of Houston for providing the data sets used in this study, and the IEEE GRSS Data Fusion Technical Committee for organizing the 2013 Data Fusion Contest.

**Conflicts of Interest:** The authors declare that they have no known competing financial interests or personal relationships that could have appeared to influence the work reported in this paper.

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
