# Peer review of "A Two-Branch Convolutional Neural Network Based on Multi-Spectral Entropy Rate Superpixel Segmentation for Hyperspectral Image Classification"

_remotesensing, doi:10.3390/rs14071569_

Round 1

Reviewer 1 Report

The authors proposed a two-branch convolutional neural network based on MERS for hyperspectral image segmentation. ERS was performed on the image of each band in an HSI. The segmentation image and the HSI were fed into the two-branch CNN. Many detailed experimental results showed that the proposed method has the highest classification accuracy. However, I still have some questions before publishing this article.

1. In the two-branch CNN, the authors used a 3D convolutional, is there any difference from a 2D convolutional that uses multiple channels? If the residual network is embedded in this structure whether it can further improve the classification effect.

2. In the proposed two-branch CNN, data fusion is performed only after the flatten layer. Why not perform data fusion at an earlier stage, where the classification accuracy might be higher?

3. As can be seen in the dataset shown in this paper, the imbalance in the sample is more severe. Therefore, it is suggested to add the analysis of the effect of unbalanced sample distribution on the classification results.

4.  In section 3.3.1, the authors designed two single-branch networks Net-X, Net-S. It needs to be shown whether they use the same hyperparameters as the proposed method.

5. In Figure 8, "(d) TBN-MERS" should be modified to "(f) TBN-MERS". It is recommended that all numbers be checked.

6. In this paper, OA, AA and Kappa were used to evaluate the classification effect. It is recommended to provide the calculation formula of these three indicators. mIoU is a very important indicator for pixel-level classification or segmentation tasks. Why not use this indicator?

7. In section 3, the author presents and discusses the results in great detail. However, the author needs to further explain the limitations and shortcomings of the proposed method.

Reviewer 2 Report

I think the authors submitted an interesting, well-written, and well-organized manuscript. The introduction and the related work sections give a very good overview about the field, the main contribution of the manuscript is clearly stated. The reviewer's concerns are the followings:

1.) The formatting of several equations are not aesthetic and weird. For example, Eq. 1 and 2. Please format the equations carefully and aesthetically in Latex. This also improves the readability of the manuscript.

2.) Two branch convolutional neural networks have been used in other image processing tasks as well. For instance, i) Composition-preserving deep approach to full-reference image quality assessment, 2020, ii) Detecting phone-related pedestrian distracted behaviours via a two-branch convolutional neural network. Please cite the related work in this respect.

3.) Since deep learning involves a lot of experiments, the publication of 
the training curves would be nice in the manuscript.

4.) I think the authors publish too little sample images about the achieved results. More images like Figure 9 would be welcomed in the manuscript.

5.) It was not clear to me that the performance numbers of other methods in Tables 9-12 come from own runnings or from the literature.

6.) The authors compare the proposed method to the state-of-the-art in Tables 9-12. In the tables, the references for SVM, 3DCNN, SSRN, HybridSN, etc. should be given. Moreover, the best and the second best results should be highlighted somehow in these tables.

7.) Are the achieved results significant? Is it possible to do somekind of significance test?

Round 2

Reviewer 1 Report

In this version, the authors have provided some additional experimental results and modifications to respond positively to my questions. However, the authors showed the training time and inference time in this version. For the same task, different computing platforms can lead to different training times and inference times. I did not find CPU or GPU information in the article. Thus, I recommend this paper to be accepted for publication after adding the hardware information.

Reviewer 2 Report

The authors answered my questions point by point. I am satisfied with the answers because the authors improved the manuscript by new experimental results. I recommend this manuscript for publication.